# Composition of Dietary Fatty Acids and Health Risks in Japanese Youths

**DOI:** 10.3390/nu13020426

**Published:** 2021-01-28

**Authors:** Masayuki Okuda, Aya Fujiwara, Satoshi Sasaki

**Affiliations:** 1Graduate School of Sciences and Technology for Innovation, Yamaguchi University, 1-1-1 Minami-Kogushi, Ube 755-8505, Japan; 2Department of Nutritional Epidemiology and Shokuiku, The National Institute of Biomedical Innovation, Health and Nutrition, 1-23-1 Toyama, Shinjuku-ku, Tokyo 162-8636, Japan; fujiwaraay@nibiohn.go.jp; 3Department of Social and Preventive Epidemiology, Graduate School of Medicine and School of Public Health, The University of Tokyo, 7-3-1 Hongo, Bunkyo-ku, Tokyo 113-0033, Japan; stssasak@m.u-tokyo.ac.jp

**Keywords:** cardiometabolic risk, compositional data analysis, Japanese, monounsaturated fatty acids, omega-3 polyunsaturated fatty acids, omega-6 polyunsaturated fatty acids, saturated fatty acids, youths

## Abstract

In the overall composition of dietary fatty acids (FAs), the quantity of each FA is interrelated with that of others. We examined the associations between dietary FA composition and cardiometabolic risk in Japanese youths. Risk factors (anthropometric characteristics, serum lipid and liver enzyme levels, and blood pressure) were measured in 5485 junior-high-school students. Dietary intake was assessed using a food frequency questionnaire. The mean saturated FA (SFA), monounsaturated FA (MUFA), omega-6 polyunsaturated FAs (PUFAs), and omega-3 PUFAs intake were 9.6%E, 10.3%E, 6.3%E, and 1.1%E, respectively. In compositional regression analysis controlled for confounders, a high intake of omega-6 PUFAs relative to others was associated with low low-density-lipoprotein cholesterol levels (LDL-C; *p* = 0.003), and relative SFA intake was associated with high levels of gamma-glutamyl transpeptidase (*p* = 0.019). Relative omega-3 PUFAs intake was associated with low blood pressure (*p* = 0.005–0.034) but had unfavorable effects on adiposity and alanine transaminase. Substitutional models showed similar results for omega-6 PUFAs on LDL-C, but MUFA had inconsistent effects on risk factors. The results from the compositional data analysis were consistent with previous studies and clinical practice/knowledge. Focusing on increasing omega-6 PUFAs in Japanese youths could have favorable consequences in the long term.

## 1. Introduction

Fat has a high-energy density and, therefore, is likely to increase total energy intake. However, evidence regarding high fat intake as the cause of obesity remains scarce. Dietary fat consists mainly of triglyceride, which is composed of one glycerol molecule with three fatty acids. Based on the number of carbon-carbon double bonds, nutritional fatty acids are classified into saturated fatty acids (SFA), monounsaturated fatty acids (MUFA), and polyunsaturated fatty acids (PUFA). PUFAs are further classified into omega-6 and omega-3 PUFAs based on the position of the first double bond from the methyl end. Several dietary guidelines focus on the effects of various fatty acids rather than on total fat intake [1,2]. A systematic review shows that health education and supplement use to increase omega-6 PUFAs intake have ameliorative health effects among children and adolescents [3]. Moreover, in a meta-analysis among adults, the iso-caloric substitution of SFA for PUFAs decreased the incidence of cardiovascular disease [4]. In fact, increasing the intake of omega-6 PUFAs is reported to have beneficial health effects in a meta-analysis that compared substitutional models between SFA and carbohydrates [5]. However, it is not possible to determine through studies that examine pair-wise substitution, e.g., SFA with omega-6 PUFAs, whether preferable health effects are associated with a decrease in SFA or an increase in PUFAs.

Compositional change in one fatty acid may influence the composition of other fatty acids in the habitual intake of food with various fat contents. Substitution models neglect this mutuality among several types of fatty acids. In order to address tangled multi-factors, compositional data analysis (CoDA) has been used in research concerned with multivariate proportion-type data, such as geochemistry to explore situations where more than two mineral elements in a compound influence each other [6]. Recently, CoDA has been introduced in the field of physical and behavioral activity to explore associations between health status and the relative proportions of physical activity, sedentary behavior, and sleeping behavior [7]. The application of CoDA in nutritional epidemiology has also been proposed [8]. CoDA can be applied to components expressed as proportions [7], and thus, can potentially be used to analyze the composition of fatty acids, which may reveal realistic associations between dietary fatty acid composition and health status.

Studies on youth intake of SFA have been conducted mostly in Europe and the United States [3]. The Japanese consume less SFA and omega-6 PUFAs and more omega-3 PUFAs than Europeans and Americans do [9]. Marine omega-3 PUFAs have protective effects against coronary heart disease [10,11]. However, the results of a meta-analysis on the effect of omega-3 PUFAs from intervention trials were inconsistent [10,12]. In adults, the environment and behaviors may intervene in the causal relationship between diet and mortality, the latter of which may result from an unfavorable lifestyle during long periods. In youths, the causality between diet and health risks may thus be simpler to interpret.

Therefore, the purpose of this study was to examine the associations between dietary fatty acid composition and cardiometabolic risk factors in Japanese junior-high-school students. We also used traditional substitution models [13] to compare the results. The hypothesis of this analysis was that SFA had unfavorable effects and omega-6 and -3 had beneficial effects on cardiometabolic risks of Japanese youths.

## 2. Materials and Methods

### 2.1. Subjects

The subjects were 5485 eighth-graders from junior-high-schools in Shunan City, Japan, who participated in the Shunan Child Cohort Study during 2006–2010. The methods of cohort selection and measurements have been described elsewhere [14,15]. Each survey was conducted between April and June. The protocol complied with the Declaration of Helsinki and was approved by the Ethics Committee of Yamaguchi University Hospital (H17-14 on 18 May 2005, H17-14-2 on 22 March 2006, H22-158 on 26 January 2011, and H22-158-1 on 22 March 2017) and the education board of Shunan City.

### 2.2. Cardiometabolic Risks

The subject’s height and weight were measured by school nurses. Body mass index (BMI) was calculated as body weight (kg)/square of body height (m) and transformed to a z-score (zBMI) using the Japanese reference in the year 2000 [16]. Blood specimens were drawn, and blood pressure was measured on a different day. The students were asked not to eat and drink anything from 10 pm on the previous day of blood draw. The levels of serum low-density lipoprotein cholesterol (LDL-C mg/dL), high-density lipoprotein cholesterol (HDL-C; mg/dL), aspartate transaminase (AST; IU/L), alanine transaminase (ALT; IU/L), and gamma-glutamyltransferase (GGT; IU/L) were measured. LDL-C, HDL-C, AST, ALT, and GGT levels were natural-log transformed because of visually-apparent skewed distributions.

### 2.3. Dietary Assessment

Dietary intake during the previous month was assessed using the brief-type self-administered diet history questionnaire for youth (BDHQ15y). The BDHQ15y is a four-page questionnaire including 64 items of food intake frequency (including food groups of cereals, meat, fish and shellfish, eggs, vegetables, fruits, nuts and pulses, potatoes, confectionaries, dairy products, fats and oils, and beverages), and 25 items of dietary habit, which the subject can answer in 15–20 min [17]. The BDHQ for an adult intake of nutrients was validated by comparing dietary records; energy-generating nutrients showed correlation coefficients of 0.27–0.64 between a single assessment of BDHQ and 16-day dietary records [18]. In BDHQ15y, the correlation coefficients were 0.22–0.48 for marine omega-3 PUFAs in red corpuscles [17] and 0.20 with urinary protein biomarker [19]. Fat, carbohydrate, and sodium were expressed as energy density, percentage of total energy intake (%E, or mg/1000 kcal only for sodium), respectively. To calculate the relative proportion (%) of SFA, MUFA, omega-6 PUFAs, and omega-3 PUFAs, their sum was considered as the total fat intake. To exclude over/under-estimation, energy intakes that were ≥0.5 times the energy expenditure of 12 to 14-year-old youths with physical activity level 1 and ≤ 1.5 times the energy expenditure of these youths with physical activity level 3 were considered plausible.

### 2.4. Confounders

Possible confounders other than zBMI were assessed in another questionnaire. These included frequency of physical activity (exercise more than twice per week), sleep duration (h), screen time (daily <1, 2, 3, 4, or ≥5 h per day), single parent (yes, or no), and the number of siblings (1, 2, or ≥3), as described elsewhere [14,15]. Ages were calculated as the difference between the blood drawing date and the birth date, divided by 365.25.

### 2.5. Statistical Analysis

We excluded subjects with (1) missing data in the BDHQ and data regarding covariates; (2) implausible energy intake; (3) physician-diagnosed heart disease, kidney disease, dyslipidemia, diabetes, or hypertension; or (4) LDL-C ≥ 140 mg/L as they may possibly have familial hyperlipidemia. Subject selection is shown in Appendix A.

Descriptive statistics for dependent variables and possible confounders were calculated as arithmetic mean ± standard deviation or counts (%). Compositional descriptive statistics including compositional geometric means, a variation matrix (dispersion), and geometric mean bar plots considering a four-part composition were calculated for SFA, MUFA, omega-6 PUFAs, and omega-3 PUFAs. Standard deviation is inappropriate for compositions because the variation of one variable is influenced by other variables. Accordingly, the pairwise covariation of fatty acids was calculated in a variation matrix. Cardiometabolic risks were categorized using quintiles to plot bar charts.

Compositional linear regression was used to examine the association between cardiometabolic risks as dependent variables and fatty acids as independent variables. Fatty acids were sequentially rotated in the order of SFA, MUFA, omega-6 PUFAs, and omega-3 PUFAs via an isometric log-ratio transformation, which created three transformed variables from four rotated settings, i.e., four sets of three transformed variables. We used the isometric log-ratio transformation with orthonormal coordinates proposed by Fiserova et al. [20]. We entered three transformed variables of fatty acids into four models for each dependent variable. *R*-squared statistics and *p* values of the models without confounders were obtained. *R*-squared statistics were used to describe the proportion of the variance explained by the fatty acid compositions, and *p* values were the results of *F*-tests on the models. Pairs of model *R*-squared and *p* values statistics were identical across the four regression models for each dependent variable.

Regression coefficients for the first-appearing part of the compositional fatty acids were obtained from the models with confounders. These coefficients indicated the effect of the first fatty acid relative to the remaining fatty acids on cardiometabolic risks. When height, weight, and zBMI were dependent variables, the confounders were age, sex, energy (kcal), carbohydrate (%E), sodium (mg/1000 kcal), physical activity, sleeping duration, screen time, single parent, and the number of siblings. When LDL-C, HDL-C, systolic and diastolic blood pressure, AST, ALT, and GGT were dependent variables, the same confounders plus zBMI were used. The effect sizes of individual fatty acids were calculated by reallocation to each fatty acid from other fatty acids in the models for cardiometabolic risks and were summarized in a change matrix. Because FAs are correlated with total fat, models including total fat (%E) instead of total carbohydrate were examined, as a sensitivity analysis. Second, we tried models including other combinations of energy, protein, fat, and carbohydrates as confounders. Third, we examined models including compositions of all energy-generating nutrients.

When analyzing traditional substitution models, a linear regression model for SFA included energy, carbohydrate, protein, MUFA, omega-3 PUFAs, and fat without fatty acids (fat minus SFA, MUFA, and omega-6 and -3 PUFAs). The variables of nutrients were expressed as energy density (%E). Possible cofounders were the same as those in the compositional regression. This model indicates the substitution of SFA for omega-6 PUFAs, which was not used in the model. Models for MUFA, and omega-6 and -3 PUFAs included energy, carbohydrate, protein, MUFA, and omega-6 and -3 PUFAs, indicating substitution of SFA.

The analysis was conducted in R 4.0.3 (R Core Team, Vienna, Austria) [21] with the software package compositions [22] and figures were produced using the packages *ggtern* [23] and ggplot2 [24]. The significance level of the test was set at <0.05.

## 3. Results

The average BMI of the subjects was 18.9 ± 2.7 kg/m^2^ in the males and 19.4 ± 2.7 kg/m^2^ in the females; the subjects with zBMI > 1.0 were 9.1% of 5485 (8.4% in the males, and 9.8% in females), and the subjects with zBMI > 2.0 were 1.1% (0.9%, and 1.3%, respectively). The proportion of the subjects with BMI >1.0 in each quintile category of cardiometabolic risks are shown in Appendix A.

The subjects’ fat consumption represented 30.1 ± 5.6% of the total energy consumption (2238 ± 640 kcal). The daily intake of FAs is shown in Table 1. The sum of fatty acids was 26.2 ± 5.1%E, which represented 87.0% of the fat intake and was composed of SFA with a geometric mean of 36.1%; MUFA with a geometric mean of 38.9%; omega-6 PUFA with a geometric mean of 20.0%; and omega-3 PUFA with a geometric mean of 4.2%. The dispersion of each fatty acid group was presented in a variation matrix (Table 2); larger co-variation between pairs indicated weaker associations. Omega-3 PUFAs showed the largest variations with other fatty acids. MUFA showed the least variations, except in one pair between omega-6 and -3 PUFAs. The distributions of the three-fatty acid sets are shown in Figure 1.

Bar plots for the quintile categories of cardiometabolic risks are shown in Figure 2. The highest categories of LDL-C, HDL-C, and GGT had the highest log-ratio of SFA and the lowest log-ratio of omega-6 PUFA. A similar tendency was seen with height and AST. However, the highest BMI category had the highest omega-3 PUFAs. Omega-6 and -3 PUFAs showed the same direction of effect on the highest risks of LDL-C, HDL-C, and GGT, but discrepant effects on blood pressure and ALT were observed.

Compositional regression analysis showed that SFA relative to other fatty acids was positively associated with GGT (*p* = 0.019), omega-6 PUFAs were negatively associated with LDL-C (*p* = 0.003), and omega-3 PUFAs were negatively associated with SBP and DBP (*p* = 0.034 and 0.005, respectively; Table 3). In contrast to having favorable effects on SBP and DBP, omega-3 PUFAs were positively associated with zBMI and ALT (*p* = 0.0485 and 0.003, respectively). Among the models with significant *F*-test statistics, height, HDL-C level, and AST level had no significant coefficients of fatty acids after adjusting for confounders. *R*-squared statistics in models with significant coefficients ranged between 0.09% for zBMI and 0.72% for GGT. In the models including total fat instead of carbohydrate, similar results were obtained (Appendix A). In models including other combinations of energy and energy-generating nutrients, the effect of omega-6 PUFAs on LDL-C did not change (Appendix A). Other effects were attenuated as the number of confounders increased, and the effect of omega-3 PUFA on zBMI was reversed in the model with full adjustment. Blood pressure was most attenuated in the models where protein intake was a confounder, whereas liver enzymes were most attenuated in the models with energy intake. zBMI was most attenuated in the models with fat intake. In models with compositions of all energy-generating nutrients, the effect of omega-6 PUFA on LDL remained significant (Appendix A). The effects of omega-3 PUFAs disappeared, but the pattern of the effects of omega-3 PUFAs in the models for fatty acids composition was similar to that of protein in the models for all energy-generating nutrients composition: negative associations with blood pressure, and positive associations with liver enzymes and adiposity.

Traditional substitution models showed that substitution of omega-6 PUFAs for SFA significantly increased log-transformed LDL-C, whereas inverse substitution significantly decreased it (both *p* < 0.001; Table 4). MUFA was positively associated with LDL-C (*p* = 0.011) but had negative associations with weight, zBMI, and ALT level (*p* ≤ 0.042).

The effect of reallocation of fatty acids in compositional regression models is shown as change matrices in Table 5. Because the mean of omega-3 PUFAs was less than 5% of the total fatty acids, a compositional 4% of each fatty acid was reallocated around the mean composition. Reallocation of FAs from SFA, MUFA, or omega-3 PUFAs to omega-6 PUFAs decreased LDL-C by 0.974–0.985 times. Reallocation of FAs from SFA, MUFA, or omega-6 PUFA to omega-3 PUFA decreased SBP by −1.610 to −1.268 mmHg and DBP by −1.711 to −1.342 mmHg. Reallocation of FAs to omega-3 increased zBMI, ALT, and GGT levels. Finally, reallocation of FAs to SFA from MUFA and omega-6 PUFAs increased GGT by 1.009–1.015 times

## 4. Discussion

The Japanese youths in this study derived 30.1% of their total daily energy intake from consumed fat. This approximates to the upper limit of a Japanese tentative dietary goal for preventing lifestyle diseases for the ages 12–14 years (20–30%E) [25]. The mean SFA intake was 9.6%E, which was less than the tentative dietary goal (10%E). Omega-6 PUFAs only exhibited a beneficial and robust effect on LDL-C in the CoDA, but unexpectedly, SFA and omega-3 PUFAs exhibited a partial and inadequate effect on cardiometabolic risks.

Previous prospective cohort and intervention studies in adults and children reveal that SFA consumption increases cardiovascular risks, whereas omega-6 PUFAs intake decreases it [26]. The beneficial effects of substituting omega-6 PUFAs for SFA could not determine which property of SFA or omega-6 PUFAs causes health modification. Traditional substitution models in this study showed that youth intake of both SFA and omega-6 PUFAs had significant but opposite associations with LDL-C. Meanwhile, compositional models showed that only high omega-6 PUFAs, relative to any other fatty acids, were significantly associated with low LDL-C. Linoleic acid, the predominant omega-6 PUFA, is metabolized to arachidonic acid, which is the precursor of inflammatory and thrombogenic cytokines. This may draw concerns regarding the recommendation of a high intake of linoleic acid [5]. However, a meta-analysis of tissue fatty acids did not find significant overall deteriorative effects of arachidonic acid [27]. The association found in this study could not explain the direct causal relationship between omega-6 PUFAs and LDL-C but indicates that focusing on promoting a diet with high omega-6 PUFAs content, instead of one with low SFA content, may help achieve low LDL-C in youths.

In previous studies of Japanese subjects, omega-3 PUFAs had a favorable effect towards reducing cardiovascular diseases [28,29]. The CoDA in this study also showed that a high intake of omega-3 PUFAs was associated with low SBP and DBP. However, dietary omega-3 PUFAs are positively associated with ALT. Japan has a lower burden of non-alcoholic fatty liver disease than other countries [30]; however, non-alcoholic hepatic steatosis is found even in Japanese youths on ultrasonographic examination [31]. Based on the results of interventional studies, supplementary omega-3 PUFAs are used to treat non-alcoholic fatty liver disease in adolescents [32] and adults [33,34]. The ALT level is a possible indicator for steatosis [35] but may not always indicate improvement because, in some studies, ALT levels remained high even after the steatosis had improved [33,36,37]. The reason for the association between omega-3 PUFAs and ALT in this study is unknown. Energy intake and zBMI, which were controlled in this association, are unlikely to be confounders. In these subjects, liver enzyme levels mostly in the normal ranges could pose concerns to explain liver dysfunction. Recent studies indicate that serum iron status is associated with liver fat or inflammation in obese youths [38,39]. Similarly in the same population of this cohort, serum iron status was associated with elevated liver enzyme levels; the levels even within the normal ranges may suggest subclinical liver dysfunction [40]. Although omega-3 PUFAs are not causally related to deteriorative hepatic function, their increased intake could not be assertively recommended for current Japanese youths based on other results. In this study, omega-3 PUFAs were associated with high BMI in the carbohydrate-adjusted compositional regression.

The overall benefit from omega-3 PUFAs was not found in Japanese youths today, who comparatively have a relatively high intake of omega-3 PUFAs in the world [9]. Regarding this point, a further increase in intake of omega -3 PUFAs could not be contradicted based on this result. Additionally, a high intake of omega-6 PUFAs might not be encouraged for European and American youths, who have already a high intake of omega-6 PUFAs. Furthermore, obesity prevalence, which is a major relevant factor to cardiometabolic risks, is lower in Japan than in other countries [41,42,43]. The optimal composition of fatty acids should be obtained through an accumulation of evidence from various dietary cultures.

GGT is elevated in liver disease and is a risk factor for coronary heart disease, but its elevated level may not exclusively be of hepatic origin [44]. In our study, SFA was significantly associated with high GGT levels in the compositional regression and with BMI and LDL-C levels in the traditional substitutional models. Therefore, it is reasonable to recommend reducing dietary consumption of SFA and increasing omega-6 PUFAs intake.

In addition to elevating LDL levels, MUFA substitution for SFA had favorable effects on low body weight, zBMI, and ALT. However, no significant associations were found in the compositional regression. MUFA is the most abundant fatty acid among the examined fatty acids, followed by SFA, but despite being an ideal exchange, the routine dietary replacement of SFA with MUFA and steady levels of other fatty acids would be difficult. MUFA has the smallest distribution range in ternary plots and the smallest covariation with SFA compared to other fatty acids. Considering the limited effects of MUFA and omega-3 PUFAs, compositional regression provides more realistic associations with health risks than traditional regression.

Although the results may be intuitively understood, the effect of the composition is vulnerable to interference from energy and energy-generating nutrients. As there were a large number of confounders, such as energy, protein, fat, or carbohydrate, the effect in the models was attenuated. The effect of omega-6 PUFAs on LDL-C was robust, but other significant effects in the energy-fat-adjusted models lessened in the fully adjusted models. However, the attenuation might be caused by an over-adjustment for energy. The composition of dietary FAs might link to and be inseparable from other nutrients in the routine diet.

There is an alternative CoDA that considers the composition of all energy-generated nutrients, such as protein, carbohydrate, SFA, MUFA, omega-6, and -3 PUFAs, and fat other than FAs [45]. The coefficients revealed a change in the health risk corresponding to a change in one component while keeping the proportions of other nutrients in the composition unchanged. It is difficult to exclusively raise or reduce one fatty acid’s proportion in a routine diet, relative to the total composition of fatty acids or energy-generating nutrients consumed.

Several limitations to this study should be considered. First, this cross-sectional study could not prove temporal causality. Nonetheless, the differences between explicable CoDA results and those of the traditional substitution models may not be caused by the cross-sectional design. Second, measuring errors in dietary assessment could affect the observed associations; however, such errors would likely lead to underestimation of the associations, not an overestimation. Third, linear associations examined in CoDA could not unveil cut-off points for health risks. Instead, ternary plots showed realistic distribution and compositions that may help understand the balance of fatty acids required in the diet. Forth, we requested the subjects fasting in blood drawing, but the compliance to complete fasting may not be ensured. Contamination of postprandial biomarkers may attenuate or obscure true associations.

## 5. Conclusions

Using CoDA, we found a significant favorable effect of omega-6 PUFA and a deteriorative effect of SFA on cardiometabolic risk factors among Japanese youths. We obtained similar results when using traditional substitutional models; however, the CoDA presented more realistic and understandable findings. Although small effect sizes for cardiometabolic risks in CoDA were seen in the *R*-squared statistics and change matrices, the favorable effects of omega-6 PUFAs may accumulate over a long period of time. Omega-3 PUFAs, however, were found to have inconsistent effects on health risks in Japanese youths. These results warrant further studies, such as a longitudinal cohort study and the examination of adult populations using CoDA.

## Figures and Tables

**Figure 1 nutrients-13-00426-f001:**
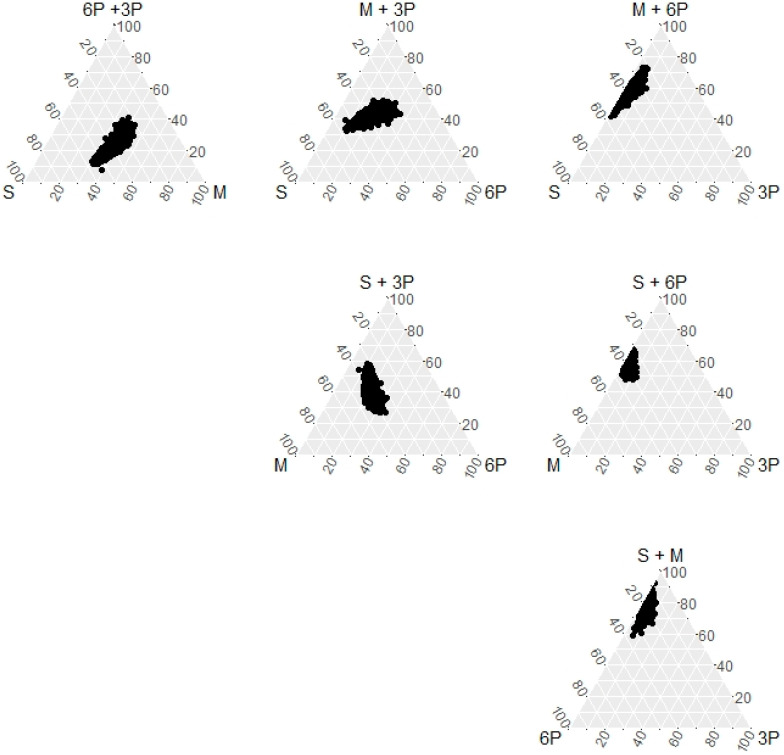
Ternary plot of fatty acids among saturated (S), monounsaturated (M), and omega-6 and -3 polyunsaturated (6P, and 3P, respectively) fatty acids. The top of the triangle indicates the sum of paired fatty acids. The scale indicates percentages of total fatty acids.

**Figure 2 nutrients-13-00426-f002:**
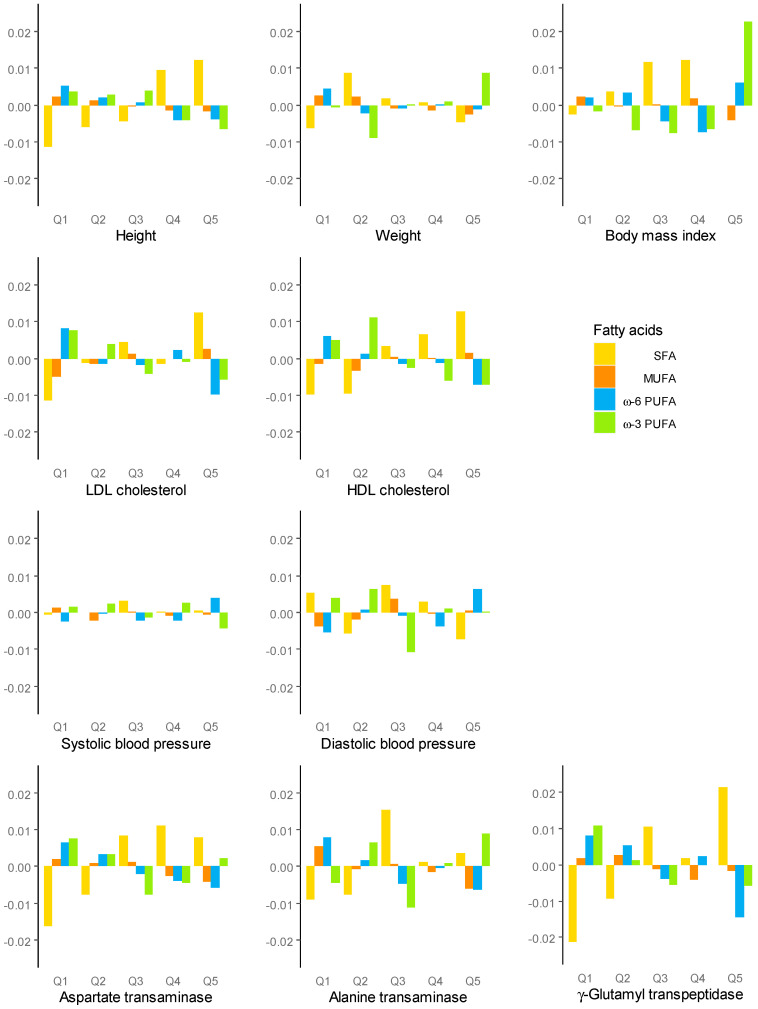
Centered log-ratios of dietary fatty acids. In each quintile category of cardiometabolic risks (Q1–Q5), a log-ratio of each geometric mean for fatty acids to overall geometric mean across all subjects, which was centered among the five categories, was plotted. A positive bar indicates a relative mean value of the fatty acid above the overall mean composition across the five categories, and a negative bar indicates a relative mean value below the overall mean.

**Table 1 nutrients-13-00426-t001:** Characteristics, health risks, and dietary intake of subjects.

	*n*	Count (%) or Mean ± SD
Sex, male		2858 (52.1)
Age, years		13.56 ± 0.29
Height, cm		156.8 ± 7.1
Weight, kg		47.2 ± 8.6
BMI, kg/m^2^		19.1 ± 2.7
LDL-C, mg/dL	4927	86.9 ± 19
HDL-C, mg/dL	4927	67.2 ± 13.8
SBP, mmHg	4944	114.5 ± 11.5
DBP, mmHg	4944	67.7 ± 8.6
AST, IU/L	4927	21.1 ± 5.9
ALT, IU/L	4927	13.5 ± 6.9
GGT, IU/L	4927	15.4 ± 5
Dietary energy, kcal		2238 ± 640
Protein, %E		14.2 ± 2.4
Fat, %E		30.1 ± 5.6
Carbohydrate, %E		54.2 ± 6.7
SFA, %E		9.6 ± 2.4
MUFA, %E		10.3 ± 2.1
PUFA, %E		6.3 ± 1.4
omega-6 PUFA, %E		5.3 ± 1.2
omega-3 PUFA, %E		1.1 ± 0.3
Sodium, mg/1000 kcal		1848 ± 429

BMI: body mass index; LDL-C and HDL-C: low- and high-density-lipoprotein cholesterol; SBP and DBP: systolic and diastolic blood pressure; AST and ALT: aspartate and alanine transaminase; GGT: gamma-glutamyl transpeptidase; SFA, MUFA, and PUFA: saturated, monounsaturated, and polyunsaturated fatty acids, respectively. *n* = 5485 for no-specific rows.

**Table 2 nutrients-13-00426-t002:** Compositional geometric means and variation matrix of fatty acids composition.

	Mean	SFA	MUFA	Omega-6 PUFAs	Omega-3 PUFAs
SFA	36.1%	0	0.031	0.077	0.112
MUFA	38.9%	0.031	0	0.018	0.046
Omega-6 PUFAs	20.0%	0.077	0.018	0	0.038
Omega-3 PUFAs	4.2%	0.112	0.046	0.038	0

SFA, MUFA, and PUFA: saturated, monounsaturated, and polyunsaturated fatty acids.

**Table 3 nutrients-13-00426-t003:** Compositional regression models.

			SFA		MUFA		Omega-6 PUFAs		Omega-3 PUFAs	
	*R*-Squared Statistics	*p*	*β* (SE)	*p*	Β (SE)	*p*	*β* (SE)	*p*	*β* (SE)	*p*
Height, cm	0.0030	<0.001	−0.22 (0.76)	0.773	0.78 (1.68)	0.642	−0.32 (1.07)	0.762	−0.06 (0.63)	0.920
Weight, kg	0.0003	0.179	−0.83 (0.98)	0.393	2.55 (2.15)	0.234	−1.52 (1.37)	0.266	0.92 (0.80)	0.253
zBMI	0.0009	0.043	0.01 (0.11)	0.896	0.10 (0.24)	0.664	−0.17 (0.15)	0.272	0.17 (0.09)	0.0495
Log (LDL-C, mg/dL)	0.0033	<0.001	0.02 (0.03)	0.496	0.08 (0.06)	0.161	−0.11 (0.04)	0.003	0.00 (0.02)	0.961
Log (HDL-C, mg/dL)	0.0016	0.012	0.03 (0.02)	0.215	−0.03 (0.05)	0.581	0.02 (0.03)	0.633	−0.02 (0.02)	0.323
SBP, mmHg	0.0001	0.330	−0.85 (1.34)	0.523	1.44 (2.95)	0.625	1.32 (1.86)	0.480	−2.35 (1.11)	0.034
DBP, mmHg	0.0010	0.046	−0.47 (1.04)	0.653	0.57 (2.28)	0.804	1.68 (1.44)	0.245	−2.41 (0.86)	0.005
Log (AST, IU/L)	0.0036	<0.001	0.01 (0.03)	0.638	−0.01 (0.06)	0.900	−0.01 (0.04)	0.710	0.02 (0.02)	0.249
Log (ALT, IU/L)	0.0048	<0.001	0.05 (0.04)	0.225	−0.10 (0.09)	0.266	−0.01 (0.06)	0.823	0.10 (0.03)	0.003
Log (GGT, IU/L)	0.0072	<0.001	0.07 (0.03)	0.019	−0.08 (0.07)	0.235	−0.01 (0.04)	0.735	0.03 (0.02)	0.182

SFA, MUFA, and PUFA: saturated, monounsaturated, and polyunsaturated fatty acids; SE: standard error; zBMI: z score of body mass index; LDL-C and HDL-C: low- and high-density-lipoprotein cholesterol; SBP and DBP: systolic and diastolic blood pressure; AST and ALT: aspartate and alanine transaminase; GGT: gamma glutamyl transpeptidase. *R*-squared statistics and their *p* values were obtained in the models of isometric log-transformed fatty acid without confounders. Regression coefficients (*β*) for the first part of the compositional fatty acids were obtained. Coefficients for height, weight, and zBMI were in the linear regression models with the following confounders: age, sex, energy (kcal), carbohydrate, sodium, physical activity, sleeping duration, screen time, single parent, and the number of siblings. Coefficients for other risks were in the linear regression models with the same confounders plus zBMI. Coefficients for risk levels correspond to an increase in the consumption of each fatty acid relative to other fatty acids.

**Table 4 nutrients-13-00426-t004:** Traditional substitution models.

	SFA ← Omega-6 PUFA		MUFA ← SFA		Omega-6 PUFAs ← SFA		Omega-3 PUFAs ← SFA	
	*β* (SE)	*p*	*β* (SE)	*p*	*β* (SE)	*p*	*β* (SE)	*p*
Height, cm	−0.045 (0.084)	0.595	−0.007 (0.136)	0.959	−0.032 (0.158)	0.839	−0.216 (0.493)	0.662
Weight, kg	−0.270 (0.107)	0.012	−0.355 (0.174)	0.042	−0.090 (0.203)	0.657	0.405 (0.631)	0.521
zBMI	−0.022 (0.012)	0.069	−0.054 (0.019)	0.005	−0.015 (0.022)	0.517	0.079 (0.070)	0.259
Log (LDL-C, mg/dL)	0.011 (0.003)	<0.001	0.012 (0.005)	0.011	−0.021 (0.006)	<0.001	−0.002 (0.018)	0.931
Log (HDL-C, mg/dL)	0.000 (0.003)	0.891	0.000 (0.004)	0.924	0.001 (0.005)	0.907	−0.029 (0.016)	0.063
SBP, mmHg	0.110 (0.147)	0.454	0.462 (0.239)	0.053	−0.012 (0.277)	0.965	−0.873 (0.866)	0.314
DBP, mmHg	0.178 (0.114)	0.118	0.385 (0.185)	0.038	0.105 (0.214)	0.625	−0.828 (0.671)	0.217
Log (AST, IU/L)	−0.004 (0.003)	0.213	−0.011 (0.005)	0.019	0.001 (0.005)	0.800	0.006 (0.017)	0.710
log(ALT, IU/L)	−0.007 (0.005)	0.114	−0.026 (0.007)	0.000	0.001 (0.009)	0.950	0.036 (0.027)	0.177
log(GGT, IU/L)	0.002 (0.003)	0.598	−0.009 (0.005)	0.088	−0.006 (0.006)	0.348	−0.002 (0.019)	0.911

Substitution of fatty acids on arrow base with fatty acids on arrow heads. SFA, MUFA, and PUFA: saturated, monounsaturated, polyunsaturated fatty acids; zBMI: z score of body mass index; LDL-C and HDL-C: low- and high-density-lipoprotein cholesterol; SBP and DBP: systolic and diastolic blood pressure; AST and ALT: aspartate and alanine transaminase; GGT: gamma-glutamyl transpeptidase; *β*: coefficient of the linear regression model; SE: standard deviation. Linear regression models for height, weight, and zBMI included the following confounders: age, sex, energy (kcal), carbohydrate, sodium, physical activity, sleeping duration, screen time, single parent, and the number of siblings. Linear regression models for other risks included the same confounders plus zBMI.

**Table 5 nutrients-13-00426-t005:** Change matrix for a 4% compositional substitution of column fatty acids with row fatty acids.

			SFA	MUFA	ω6-PUFAs	ω3-PUFAs
Height, cm	diff.	SFA	0	−0.075	0.042	0.135
		MUFA	0.072	0	0.112	0.205
		ω6-PUFAs	−0.029	−0.106	0	0.104
		ω3-PUFAs	−0.014	−0.091	0.026	0
Weight, kg	diff.	SFA	0	−0.192	0.215	−2.426
		MUFA	0.189	0	0.395	−2.246
		ω6-PUFAs	−0.155	−0.356	0	−2.589
		ω3-PUFAs	0.619	0.418	0.825	0
zBMI	diff.	SFA	0	0.005	0.033	−0.442
		MUFA	−0.005	0	0.028	−0.447
		ω6-PUFAs	−0.028	−0.022	0	−0.470
		ω3-PUFAs	0.100	0.105	0.133	0
LDL-C, mg/dL	ratio	SFA	1	0.993	1.023	1.004
		MUFA	1.006	1	1.030	1.011
		ω6-PUFAs	0.981	0.974	1	0.985
		ω3-PUFAs	0.997	0.991	1.021	1
HDL-C, mg/dL	ratio	SFA	1	1.005	1.000	1.054
		MUFA	0.995	1	0.995	1.049
		ω6-PUFAs	0.999	1.005	1	1.053
		ω3-PUFAs	0.986	0.991	0.986	1
SBP, mmHg	diff.	SFA	0	−0.244	−0.334	5.813
		MUFA	0.236	0	−0.106	6.041
		ω6-PUFAs	0.295	0.043	0	6.100
		ω3-PUFAs	−1.268	−1.521	−1.610	0
DBP, mmHg	diff.	SFA	0	−0.153	−0.364	6.009
		MUFA	0.147	0	−0.222	6.151
		ω6-PUFAs	0.310	0.153	0	6.315
		ω3-PUFAs	−1.342	−1.500	−1.711	0
AST, IU/L	ratio	SFA	1	1.003	1.004	0.941
		MUFA	0.997	1	1.001	0.938
		ω6-PUFAs	0.997	1.000	1	0.938
		ω3-PUFAs	1.013	1.017	1.017	1
ALT, IU/L	ratio	SFA	1	1.018	1.007	0.781
		MUFA	0.983	1	0.991	0.768
		ω6-PUFAs	0.993	1.011	1	0.776
		ω3-PUFAs	1.054	1.073	1.062	1
GGT, IU/L	ratio	SFA	1	1.015	1.009	0.926
		MUFA	0.985	1	0.995	0.913
		ω6-PUFAs	0.991	1.006	1	0.918
		ω3-PUFAs	1.012	1.028	1.022	1

Change of risk factors for differences (diff.) or ratios due to changes in fatty acid composition in the linear regression models shown in Table 3. SFA, MUFA, and PUFA: saturated, monounsaturated, and polyunsaturated fatty acids; zBMI: z score of body mass index; LDL-C and HDL-C: low- and high-density-lipoprotein cholesterol; SBP and DBP: systolic and diastolic blood pressure; AST and ALT: aspartate and alanine transaminase; GGT: gamma-glutamyl transpeptidase.

## Data Availability

The data presented in this study are available on request from the corresponding author.

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
