# Peer review of "Composition of Dietary Fatty Acids and Health Risks in Japanese Youths"

_nutrients, 2021, doi:10.3390/nu13020426_

Round 1

Reviewer 1 Report

The main variables of the diet should be better explained (fatty acid intake).

Describe Compositional data analysis (CoDA) and BDHQ.

The claim that the consumption of omega 3 PUFA and ALT are related may have biases in its cause, such as the increase in BMI. The ALT values are within normality and the association with non-alcoholic hepatic steatosis does not seem justified. Statements made about ALT and omega-3s should be written differently.

Author Response

Thank you for your instructive comments. They have helped us to improve the manuscript for readers. We have added more explanatory sentences according to the reviewer‘s comments and corrected some mistyping words in red fonts.

Comment 1. The main variables of the diet should be better explained (fatty acid intake).

Response 1. Thank you for your comments. We have added sentences about fat and fatty acids in the Introduction.

Dietary fat consists mainly of triglyceride, which is composed of one glycerol molecule with three fatty acids. Based on the number of carbon-carbon double bonds, nutritional fatty acids are classified into saturated fatty acids (SFA), monounsaturated fatty acids (MUFA), and polyunsaturated fatty acids (PUFA). PUFAs are further classified into omega-6, and omega-3 PUFAs based on the position of the first double bond from the methyl end.” Lines 36–41.

Comment 2. Describe Compositional data analysis (CoDA) and BDHQ.

Response 2. Thank you. We have added explanations as the following;
Compositional data analysis (CoDA) in the Introduction
In order to address tangled multi-factors, compositional data analysis (CoDA) has been used in research concerned with multivariate proportion-type data, such as geochemistry to explore situations where more than two mineral elements in a compound influence each other.“ Lines 52–55.
BDHQ in the Methods
The BDHQ15y is a four-page questionnaire including 64 items of food intake frequency (including food groups of cereals, meat, fish and shellfish, eggs, vegetables, fruits, nuts and pulses, potatoes, confectionaries, dairy products, fats and oils, and beverages), and 25 items of dietary habit, which the subject can answer in 15–20 min [17]. “ Lines 95–98.

Comment 3. The claim that the consumption of omega 3 PUFA and ALT are related may have biases in its cause, such as the increase in BMI. The ALT values are within normality and the association with non-alcoholic hepatic steatosis does not seem justified. Statements made about ALT and omega-3s should be written differently.

Response 3. zBMI and energy intake were adjusted in linear regression models for liver enzymes. It is a challenging issue whether this cohort had NAFLD with elevated liver enzyme levels. We previously reported that elevated enzyme levels in this cohort, even with liver enzymes almost within normal ranges, were associated with serum iron status; these associations are similar in recent research, which showed iron status was associated NAFLD in obese children and adolescents. We have added the following explanations in the Discussion.

Energy intake and zBMI, which were controlled in this association, are unlikely to be confounders. In these subjects, liver enzyme levels mostly in the normal ranges could pose concerns to explain liver dysfunction. Recent studies indicate that serum iron status is associated with liver fat or inflammation in obese youths [38, 39]. Similarly in the same population of this cohort, serum iron status was associated with elevated liver enzyme levels; the levels even within the normal ranges may suggest subclinical liver dysfunction [40].” Lines 282–287.

Added References

  1. Zhang, J.; Cao, J.; Xu, H.; Dong, G.; Huang, K.; Wu, W.; Ye, J.; Fu, J. Ferritin as a key risk factor for noalcoholic fatty liver disease in children with obesity. J Clin Lab Anal 2020, 00:e23602, doi: 10.1002/jcla.23602
  2. Mörwald, K.; Aigner, E.; Bergsten, P.; Brunner, S.M.; Forslund, A.; Kullberg, J.; Ahlström, H.; Manell, H.; Roomp, K.; Schütz, S.; Zsoldos, F.; Renner, W.; Furthner, D.; Maruszczak, K.; Zandanell, S.; Weghuber, D.; Mangge, H.Serum Ferritin Correlates With liver fat in male adolescents with obesity. Front Endocrinol 2020, 11, 340, doi: 10.3389/fendo.2020.00340.
  3. Okuda, M.; Sasaki, S.; Kunitsugu, I.; Sakurai, R.; Yoshitake, N.; Hobara, T. Iron load and liver enzymes in 10- and 13-year-olds. J Pediatr Gastroenterol Nutr 2011,52, 333-338, doi: 10.1097/MPG.0b013e318201aecc.

Reviewer 2 Report

“Composition of dietary fatty acids and health risks in Japanese adolescents” by coauthors Okuda, Fujiwara, and Sasaki, presents nutritional data from a large cohort of junior high school students. Coauthors used a statistical approach called compositional data analysis to analyze the data set, with results indicating associations between fatty acid groups and cardiometabolic biomarkers. The manuscript overall is well-written. The introduction explains the rationale for compositional analysis very well, and identifies the complexities of interpretation when alternate methods of analysis are used. The figures and tables are clear with appropriate labeling. In particular, the ternery plots were illustrative. Other strengths include the large cohort, methodological justification provided, the explanation of the CoDA statistical approach, and consistency of results with published literature.

Specific Comments for clarification/improvement:

The term “adolescents” in the title suggests that the cohort were teenagers, or in puberty. However, the mean age was 13.56 with SD 0.29, and pubertal status is highly variable in this age group, especially in males vs females. Perhaps describing the cohort as “youth” might be more accurate.

The objective is clear in the introduction, however this section would benefit from a statement of hypothesis. Did the authors predict that omega 6 (instead of omega 3) would demonstrate the associations?

Please comment on whether the laboratory tests were drawn while fasting. Non-fasting status would affect the triglyceride levels, but would not impact the total cholesterol, LDL, or HDL measurements.  

Regarding the centered log-ratios of FAs in each quintile category of risk factors, can the authors comment on whether any of the quintiles corresponded to overweight/obesity in this cohort based on 85% for age and gender? This would give the reader a sense of the health status of the cohort. 

The introduction points out that the Japanese population consumes more omega 3 FA than populations in other countries. This is an important point because Western diets often have disproportionately high omega 6 intake relative to omega 3 (15:1). Therefore, Western countries, in contrast, tend to promote omega 3 intake.  It would be helpful to circle back to this point in the discussion to provide an international context for the findings.

Regarding the discussion section about NAFLD, ALT is known not to correlate with severity of hepatic steatosis or steatohepatitis, although it is useful as an indicator to evaluate someone for NAFLD. Omega 3 PUFAs have been studied (as the authors noted), but are currently not approved pharmaceutical interventions for NAFLD. In assessing the applicability of this discussion, can the authors comment on the possibility of NAFLD in this study cohort based on the reported AST and ALT results? Again, this would give some perspective on health status of the cohort. 

Author Response

Thank you for your comments and advices. These comments are extremely helpful for us to improve this manuscript for readers. We have added some sentences according to the reviewer’s comments and corrected some mistyping words in red fonts.

Comment 1. The term “adolescents” in the title suggests that the cohort were teenagers, or in puberty. However, the mean age was 13.56 with SD 0.29, and pubertal status is highly variable in this age group, especially in males vs females. Perhaps describing the cohort as “youth” might be more accurate.

Response 1. Thank you for your comment. We have replaced all “adolescent” with “youth” throughout the text including the title.

Comment 2. The objective is clear in the introduction, however this section would benefit from a statement of hypothesis. Did the authors predict that omega 6 (instead of omega 3) would demonstrate the associations?

Response 2. We have added the hypotheses in the Introduction, and main conclusion in the head of the Discussion.

The hypothesis of this analysis was that SFA had unfavorable effects and omega-6 and -3 had beneficial effects on cardiometabolic risks of Japanese youths.” Lines 71–73

Omega-6 PUFAs exhibited only beneficial and robust effect on LDL-C in the CoDA, but unexpectedly, SFA and omega-3 PUFAs exhibited partial and inadequate effect on cardiometabolic risks.” Lines 255–257.

Comment 3. Please comment on whether the laboratory tests were drawn while fasting. Non-fasting status would affect the triglyceride levels, but would not impact the total cholesterol, LDL, or HDL measurements.  

Response 3. Thank you for your comment. In accordance with you comments, we have added the following sentences in the Methods and Discussion. However, the TG level may be influence by taking breakfast before blood drawing, and we could not ensure the compliance to complete fasting. Therefore, we deleted methods and results related to TG.

The students were asked not to eat and drink anything from 10 pm on the previous day of blood drawing.” Lines 87–88.

Forth, we requested the subjects fasting in blood drawing, but the compliance to complete fasting may not be ensured. Contamination of postprandial biomarkers may attenuate or obscure true associations.” Lines 330–332.

Comment 4. Regarding the centered log-ratios of FAs in each quintile category of risk factors, can the authors comment on whether any of the quintiles corresponded to overweight/obesity in this cohort based on 85% for age and gender? This would give the reader a sense of the health status of the cohort. 

Response 4. Obesity and overweight prevalence in Japan are low in the reports from GBD, OECD, and WHO. These references are relevant to the next response. We have calculated BMI and zBMI, described the proportion of zBMI >1.0, and >2.0, and added Supplementary Table 3.

Average BMI of the subjects was 18.9 ± 2.7 kg/m2 in the males and 19.4 ± 2.7 kg/m2 in the females; the subjects with zBMI >1.0 were 9.1% of 5485 (8.4% in the males, and 9.8% in females), and the subjects with zBMI >2.0 were 1.1% (0.9%, and 1.3%, respectively). The proportion of the subjects with BMI >1.0 in each quintile category of cardiometabolic risks were shown in Supplementary Table S3. “ Lines162–166.

Added References

  1. The GBD 2015 Obesity Collaborators. Health effects of overweight and obesity in 195 countries over 25 years. N Engl J Med 2017; 377:13-27. doi: 10.1056/NEJMoa1614362.
  2. World Health Organization. The Global Health Observatory. World Health Organization, Geneva, 2017. https://www.who.int/data/gho. (Accessed on January 24th 2021).
  3. The Heavy Burden of Obesity: The Economics of Prevention, OECD Health Policy Studies. OECD Publishing, Paris. 2020. doi: 10.1787/67450d67-en.

Comment 5. The introduction points out that the Japanese population consumes more omega 3 FA than populations in other countries. This is an important point because Western diets often have disproportionately high omega 6 intake relative to omega 3 (15:1). Therefore, Western countries, in contrast, tend to promote omega 3 intake.  It would be helpful to circle back to this point in the discussion to provide an international context for the findings.

Response 5. Thank you for your comment. We have added sentences to explain these results cannot be simply adopted to people in other countries, who take fat with a different fatty acids composition.

Overall benefit from omega-3 PUFAs was not found in Japanese youths today, who had relatively high intake of omega-3 PUFAs in the world [9]. Regarding this point, further increase in intake of omega -3 PUFAs could not be contradicted only from this result. Either, high intake of omega-6 PUFAs might not be encouraged to European and American youths, who have already high intake of omega-6 PUFAs. In addition, obesity prevalence, which is a major relevant factor to cardiometabolic risks, is lower in Japan than in other countries [41, 42, 43]. Optimal composition of fatty acids should be obtained through an accumulation of evidence from various dietary cultures. Lines 291–297.

Comment 6. Regarding the discussion section about NAFLD, ALT is known not to correlate with severity of hepatic steatosis or steatohepatitis, although it is useful as an indicator to evaluate someone for NAFLD. Omega 3 PUFAs have been studied (as the authors noted), but are currently not approved pharmaceutical interventions for NAFLD. In assessing the applicability of this discussion, can the authors comment on the possibility of NAFLD in this study cohort based on the reported AST and ALT results? Again, this would give some perspective on health status of the cohort. 

Response 6. It is a challenging issue whether this cohort had NAFLD with elevated liver enzyme levels. We previously reported that elevated enzyme levels in this cohort, even with liver enzymes almost within normal ranges, were associated with serum iron status; these associations are similar in recent research, which showed iron status was associated NAFLD in obese children and adolescents. We have added the following explanations in the Discussion.

In these subjects, liver enzyme levels mostly in the normal ranges could pose concerns to explain liver dysfunction. Recent studies indicate that serum iron status is associated with liver fat or inflammation in obese youths [38, 39]. Similarly in the same population of this cohort, serum iron status was associated with elevated liver enzyme levels; the levels even within the normal ranges may suggest subclinical liver dysfunction [40].” Lines 282–287.

Added References

  1. Zhang, J.; Cao, J.; Xu, H.; Dong, G.; Huang, K.; Wu, W.; Ye, J.; Fu, J. Ferritin as a key risk factor for noalcoholic fatty liver disease in children with obesity. J Clin Lab Anal 2020, 00:e23602, doi: 10.1002/jcla.23602
  2. Mörwald, K.; Aigner, E.; Bergsten, P.; Brunner, S.M.; Forslund, A.; Kullberg, J.; Ahlström, H.; Manell, H.; Roomp, K.; Schütz, S.; Zsoldos, F.; Renner, W.; Furthner, D.; Maruszczak, K.; Zandanell, S.; Weghuber, D.; Mangge, H.Serum Ferritin Correlates With liver fat in male adolescents with obesity. Front Endocrinol 2020, 11, 340, doi: 10.3389/fendo.2020.00340.
  3. Okuda, M.; Sasaki, S.; Kunitsugu, I.; Sakurai, R.; Yoshitake, N.; Hobara, T. Iron load and liver enzymes in 10- and 13-year-olds. J Pediatr Gastroenterol Nutr 2011,52, 333-338, doi: 10.1097/MPG.0b013e318201aecc.
